# Influences of Adult Gender and Parenthood on Adult-Child Interaction Style

**DOI:** 10.3390/children9121804

**Published:** 2022-11-24

**Authors:** Darcy K. Smith, Ran An, Klaus Libertus

**Affiliations:** Department of Psychology, University of Pittsburgh, Pittsburgh, PA 15260, USA

**Keywords:** parent–child interaction, language development, gender, child-directed speech, fathers

## Abstract

Previous studies report differences between mothers and fathers during parent–child interactions. However, the origins of these differences remain unknown. We address this gap by examining the impact of adult gender and gender perceptions on adult-child interactions. Unlike previous studies, we observed both parent and non-parent adults during one-on-one interactions with a child. Further, for non-parent adults the child’s identity was held constant while the child’s assumed gender was actively manipulated using clothing cues. Results reveal systematic differences between parents and non-parents, but also between male and female adults in language quantity, quality, and engagement strategies during adult-child interactions. Adults’ perceptions of gender roles partially explain these findings. In contrast, the child’s gender did not impact adult-child interactions. Together, our results support the notion that male and female adults offer unique contributions to a child’s development.

## 1. Introduction

Parents are important for children’s early development by providing physical, emotional, and intellectual stimulation and support. Especially during parent–child play, parents offer learning opportunities that are critical for the child’s acquisition of new skills across domains [1]. For example, asking parents to engage their child in a new motor skill fosters the acquisition of this motor skill [2,3]. Further, research demonstrates that such parent-guided experiences have long-lasting implications bridging developmental domains [4,5]. However, the child is not merely a passive recipient of stimulation during play but is actively shaping the interaction between parent and child. Specifically, child behavior directly influences parent behavior during social exchanges [6,7]. In short, both the parent and the child determine how parent–child interactions unfold. This observation also suggests that both factors related to the parent and factors related to the child impact parent–child interaction style and quality. Child-related factors include differences in temperaments, abilities, and interests. Adult- or parent-related factors include varying levels of experience with children, distinct roles in the family context, and particular beliefs about parenting. To determine the unique contributions of adult-related factors during adult-child interactions, one would need to control for influences stemming from child-related factors. The current study aims to address this issue by examining adult-child interactions while controlling for child-related factors by keeping the child constant (i.e., using the *same* child) across a group of non-parent adult participants. This innovative approach allows us to systematically examine what adult factors (e.g., gender, experiences, perceptions, or interests) influence how adults interact with children. To further examine the effect of parenthood on parent–child interactions, we also include a group of female parents who engage with their own child. This comparison between non-parent males, non-parent females, and parent females allows us to analyze what aspects of adult-child interaction quality and quantity are related to adult gender, parenthood status, and child gender.

The importance of adult-child interactions may be best understood in the domain of language development. How much parents communicate with their children during interactions varies significantly both within and between parents from different socio-economic backgrounds [8,9]. Evidence suggests that differences in child-directed communication impact the child’s development. For example, the amount of language directed toward the child during parent–child interactions predicts children’s subsequent language learning [10]. However, beyond the sheer quantity of speech, language quality matters as well. For example, parents who talk more also tend to use more complex language [11]. Further, parental education, age, and gender also impact language quality [12,13].

It has long been hypothesized that fathers interact and stimulate children differently than mothers–potentially due to differences in the amount of experience fathers have with their children compared to mothers [14]. Arguing against this hypothesis, some studies reported no differences between mothers and fathers in their language engagement directed toward their infants [15], their language modification and simplification [16], or their overall responsiveness toward the child [17]. In contrast, more recent research has identified differences between male and female parents (note: we will refer to female parents as mothers and male parents as fathers in the following). For example, mothers use more soothing techniques during interactions and seem overall more responsive than fathers [18,19]. At the same time, fathers direct more wh-questions toward the child than mothers [12,20,21]. These differences are noteworthy and may have developmental consequences. For example, asking wh-questions challenges the child and may encourage the development of subsequent reasoning skills, receptive vocabulary, and a child’s own production of wh-questions [22,23]. Together, these findings suggest that mothers and fathers uniquely contribute to the child’s learning and development through different interaction styles.

Understanding why mothers and fathers interact differently with their children remains unknown. This gap in knowledge is partly due to a lack of fathers in developmental research [24]. Fathers may be less likely to participate in research because they perceive research opportunities as being directed primarily toward mothers [25]. In addition to such a relative lack of studies on fathers, several confounding factors also impede our understanding of how mothers and fathers differ. For example, disparities in childcare experiences, family roles, and familiarity with the child affect parent behavior but are orthogonal to the adult’s gender. In fact, going beyond parents, the differences in adult-child interactions between females (mothers) and males (fathers) remain a poorly understood area in general. Therefore, the presence of gender-related differences between females and males when interacting with children remains an open question.

The current study aims to address the gap in our understanding of adult-child interactions across adult genders and roles by examining interactions between *non-parent* undergraduate students and an unrelated child as well as interactions between a female parent (mother) and their own child in a standardized playroom. Within our adult non-parent sample, the current study design is innovative by holding the child constant. This approach controls for both factors associated with the child (e.g., temperament, familiarity with primary caregiver) and factors related to the caregiver role (i.e., primary vs. secondary caregiver). In addition, this study manipulates the perceived gender of the child by varying the child’s clothing. We hypothesized that previously reported differences in adult-child interaction styles between mothers and fathers will replicate in a sample of non-parent undergraduate students. Specifically, we predicted that females would direct more speech toward the child (quantity), but males would use more complex language such as wh-questions (quality). However, we predicted that female parents would show the highest levels of engagement quantity and quality when compared to non-parents. That is, we predict higher levels of similarity between non-parent males and females compared to parent females across all of our measures. A final innovation of the current study is an attempt to go beyond gender as a binary construct. Specifically, the current study examines the impact of individual differences in adults’ perceptions regarding gender roles or interest in gendered activities on adult-child language engagement. To quantify individual differences in broad gender perceptions, we introduce the Experiences, Perceptions, and Interests Questionnaire (EPIQ, see Appendix A).

## 2. Materials and Methods

A local Internal Review Board (IRB) approved all study materials and procedures to ensure compliance with ethical research standards. Adult participants provided written informed consent before completing any study-related activities. A parent of the child participant provided written informed consent before each study session. Following all study procedures, adult participants were debriefed about the study. All recruitment and data collection activities were completed between December 2018 and February 2020. 

### 2.1. Participants

For this study, two waves of recruitment were conducted. The first wave included a total of 91 adults and one toddler. All adults were undergraduate students enrolled in an Introduction to Psychology class and received course credit for their participation. None of the adults reported having a child of their own. Six scheduled adult participants missed their appointments. Of the remaining 85 adults, three males were removed before analysis due to their total word counts being more than two standard deviations above the group mean. No such extreme observations were present in the female participants. The second wave of recruitment included a total of 25 female parents and their toddlers (toddlers 29% female, *M*_age_ = 16.85 months, *SD*_age_ = 0.70). All parent–child dyads received a small toy or board book for their participation. One parent missed her appointment. The final sample consisted of 106 adults (65 females). Since this is the first study to examine adult-child interactions between undergraduate students and an unfamiliar child, no a priori estimates of effect sizes were available. However, the sample included here is similar in size to prior studies examining adult-child communication in fathers (80 participants) [12] and adult-child interactions in mothers and fathers (33 mothers vs. 33 fathers) [21]. Participant demographics are representative of the local population of undergraduate students (see Table 1). Unfortunately, adults’ age was not recorded as part of the current study. However, 94% of the undergraduate population at the institution is 24 years of age or younger. Further, since all students were enrolled in Introduction to Psychology, we estimate that most adults were in their first year of study and around 20 years of age.

The child participating in the first wave of this study was one Caucasian female toddler, aged 14.5 months at study onset and 18 months at study offset (*M* = 16.61 months, *SD* = 0.93). The age of the toddler increased over the course of the study but was in between 16.5 and 17.5 months for most observations (*n* = 42).

### 2.2. Procedure and Measures

Non-parent adult participants completed one self-report questionnaire and a 15 min in-lab adult-child interaction (see below). Whenever possible, two testing sessions were scheduled back-to-back to make the best use of the toddler’s awake and alert time while in the lab. To accommodate this back-to-back schedule, about half of the adults completed the questionnaire after their interaction with the child, and the other half completed the questionnaire beforehand. A brief warm-up period occurred before the adult-child interaction where the child was introduced as “Sammy” who was either dressed in “boyish” clothing or “girlish” clothing (e.g., blue sweater vs. pink sweater).

Parent participants were given the same self-report questionnaire and participated in a 15 min in-lab parent–child interaction in the same location with identical toys to those used in the non-parent adult-child interaction. However, parent participants interacted with their own child. Additionally, all parents completed the self-report questionnaire after the parent–child interaction to reduce the potential for child fussiness.

### 2.3. Adult-Child Interactions

Adults (parent or non-parent) completed a 15 min one-on-one interaction with their child (for parents) or an unknown toddler (non-parents). For non-parents, the child’s gender was unknown and could only be guessed based on the child’s clothing. Further, for non-parents the actual father of the child was always present in the room but pretending to read a book and did not engage with the child during the interaction (except for instances where the child was crying and upset). All adult-child interactions were completed in a standardized rectangular room containing a variety of toys, books, and images on the walls. Selected toys included dolls, stuffed animals, blocks, balls, figurines, trucks, and mechanical toys, such as a ring stacker or ball chute toys. The number of toys chosen were balanced between stereotypically male (e.g., trucks), female (e.g., dolls), and neutral toys (e.g., blocks). All toys were set up in a consistent arrangement around the perimeter of the room across all adult participants. New toys were introduced on 11 random occasions during the semester to maintain the interest of the toddler participating with the non-parent groups (as this was the same toddler throughout the experiment). Adults were instructed: “We would like to see how you play with [child’s name] for 15 min. You can use all the toys and objects in this room.” Non-parent adults were further instructed: “Sammy’s dad will be in the room but will read a book. We want to see how you play with Sammy.” All adult-child interactions were video recorded using three wall-mounted cameras providing adequate coverage of the room during free-play exchanges. After 15 min of free play, non-parent adults were debriefed on the deception regarding the toddler’s true gender. Parents were given a tablet to complete the self-report questionnaire while a research assistant continued playing with their child. 

### 2.4. Experiences, Perception, and Interests Questionnaire (EPIQ) 

To measure the strength of gender and child-related experiences, we collated two measures of gendered interests and perceptions and created a basic measure of child-centered experiences. Questions about gender perceptions, attitudes, and gendered interests were adapted from two previous studies [26,27]. The resulting 43-item Experiences, Perceptions, and Interests Questionnaire (EPIQ) was used to quantify participants’ experiences of caring for children (11 questions), perceptions toward gender stereotypes and roles (22 questions), and interests in gendered activities (10 questions). To confirm the utility of the EPIQ as a measure of gender-related factors beyond the male-female dichotomous gender categorization, an additional sample of 77 undergraduate students (27 female, one other) and 73 parents (69 female, four female) completed only the EPIQ (data collected in December 2018). Results from this initial sample were used for a Structural Equation Modeling analysis to determine whether experiences, perceptions, and interests supported a dichotomous male-female latent variable model. The model shows poor fit, χ2(779, N = 256) = 1923.43, *p* < 0.001, TLI = 0.62, CFI = 0.60; suggesting that experiences with children, perceptions of gender stereotypes, and interests in gendered activities capture gendered factors as a spectrum rather than as a dichotomous measure of gender. To examine the impact of gender-related factors on adult-child interactions, we further calculated standardized scores for the Experiences, Perceptions, and Interests factors for each participant (standard scores were designed with a mean of 50 and based on all collected EPIQ responses). Higher scores on each factor indicate more experiences with children (Experience), more egalitarian perceptions about male and female roles (Perception), and stronger interest in activities potentially considered as more feminine (Interests). The EPIQ may be of interest to for future research and is available for free (see Appendix A).

### 2.5. Analyses

All adult-child interactions were fully transcribed by trained coders and each transcription was checked by a second coder for accuracy. Any disagreements were resolved either by the second coder or during weekly coding meetings. The toddlers were pre-verbal at the time of the study and offered only a few sounds and utterances during the interaction. Therefore, only adult language quantity and content were examined. Transcriptions of adult speech were processed using custom MATLAB scripts to quantify the number of tokens uttered per minute during the interaction and the proportion of types used from seven broad categories: Noun, verb, number, space, question, color, and emotion words (see Table 2 for examples). Short words (1 character only), long words (>15 characters), and sounds or other non-words were excluded prior to analysis. Finally, adult engagement styles during adult-child interactions were coded using the Parenting Interactions with Children Checklist of Observations Linked to Outcomes (PICCOLO) [28]. To check for agreement, 15 non-parent and five parent interactions were double coded by a second trained coder. Interrater reliability for the PICCOLO ratings was high (r = 0.91).

## 3. Results

### 3.1. Experiences, Perceptions, and Interests Questionnaire (EPIQ)

In the current study, the adult’s (self-reported) gender is a critical grouping variable and hypothesized source of differences in adult-child interaction styles. However, we do acknowledge that constructing gender as a dichotomy is an oversimplification. Therefore, we wanted to first examine whether scores on the Experiences, Perceptions, and Interests Questionnaire (EPIQ) may be used as an alternate variable that reflects aspects of gender-perceptions in our adult participants. A Multivariate Analysis of Variance (MANOVA) was used to compare our three groups of Interaction Partners: non-parent males (NPM), non-parent females (NPF), and parent females (PF). Interaction partners were compared on the three dimensions of the EPIQ (Experiences, Perception, and Interests). Results reveal a significant effect of Interaction Partner, F(6,206) = 15.05, *p* < 0.001, indicating a difference in gender-related factors between groups. Univariate follow-up ANOVAs indicate a significant difference between interaction partner groups for Perception, F(2,104) = 67.61, *p* < 0.001, Interactions F(2,104) = 16.29, *p* < 0.001, and Experiences, F(2,104) = 5.75, *p* = 0.004. Follow-up Tukey multiple comparisons of means tests suggest that male participants differ from female participants on the Perceptions (*ps* < 0.001) and Interests (*ps* < 0.001) sub-scales of the EMQ (regardless of parenthood status). In contrast, follow-up comparisons for Experiences showed no significant difference between non-parents (*p* = 0.171), or between females (*p* = 0.165), but a significant difference between NPM and PF (*p* = 0.003). Differences on the Perception and Interest factors are theoretically predicted and indicate that the EPIQ provides a continuous measure of gender-related differences. Therefore, we will examine the impact of the Perception and Interest factors on adult-child interactions. In contrast, the Experiences factor will be treated as a confounding factor and will be statistically controlled for in our analyses.

### 3.2. Language Quantity

A 3 (Interaction Partner) by 2 (Toddler Gender) between-subjects Analysis of Variance (ANOVA) was used to determine the overall quantity of language used by each parent and non-parent adult. To control for effects of participants’ experience with children, this analysis was repeated including adults’ EPIQ Experience scores as a covariate in the model. We hypothesized that female participants would produce more words during the interaction than male participants. Our results support this hypothesis. The ANCOVA reveals only a marginally significant main effect of Interaction Partner, F(2,104) = 3.082, *p* = 0.0502, no main effect of Toddler Gender, F(1,102) = 0.007, *p* = 0.931, and no significant interaction, F(2,104) = 1.22, *p* = 0.298. However, the effect of Interaction Partner reaches statistical significance when controlling experience with children, F(2,104) = 3.09, *p* = 0.050, suggesting that differences in overall quantity of language between parent and non-parent male and female adults may be explained by differences in experience with children and gender-related factors between groups. These results are illustrated in Figure 1 and indicate that parents, female adults, and adults with higher levels of experience with young children engage in more talking during adult-child interactions.

### 3.3. Language Quality

In addition to differences in the number of words used (language quantity), we also examined differences in language quality (i.e., what kinds of words are used) between interaction partner groups. For this analysis, we calculated the proportions of seven broad word-types (see Table 2). Based on previous research, we predicted that male adults would use wh-question words more frequently than female adults. A 3 (Interaction Partner) by 2 (Toddler Gender) Multivariate Analysis of Co-Variance (MANCOVA) including the EPIQ factor Experiences as a covariate revealed a significant main effect of Interaction Partner, F(14, 190) = 3.82, *p* < 0.001, and Experiences, F(7, 94) = 3.03, *p* = 0.006, but no effect of Toddler Gender (*p* = 0.428), and no interaction (*p* = 0.481). Separate follow-up ANCOVAs for each word category revealed significant differences between interaction partner groups in three categories. NPM used a higher proportion of wh-question words, F(2, 104) = 5.90 *p* = 0.004. In contrast, FP used a higher proportion of nouns, F(2, 104) = 10.17, *p* < 0.001 and a lower proportion of emotion words, F(2, 104) = 10.24, *p* < 0.001 compared to NPM and NPF (see Figure 2), suggesting that parents spend less time using emotion words and more time labeling during interactions in comparison to non-parents. We observe no effects for all other word categories (all *ps* > 0.102). These results reveal systematic differences in language quality between both gender and parenthood status when playing with a young child.

### 3.4. Engagement Strategies

Finally, we examined whether differences in engagement styles (as assessed by the PICCOLO) were evident between interaction partner groups during interactions with a toddler. We predicted that parents would engage in more diverse strategies than less-experienced non-parent males and females. Analyses indicate, contrary to our hypothesis, a divide by gender rather than by parenthood status. A Multivariate Analysis of Variance (MANOVA) was used to compare interaction partner groups on four engagement strategies: Affection, Responsiveness, Encouragement, and Teaching. The multivariate result was significant for interaction partner, F(8,204) = 6.02, *p* < 0.001, indicating a difference in engagement strategies between interaction partner groups (see Figure 3). The univariate F tests showed there was no significant difference between interaction partner groups for Affection, F(2,104) = 1.20, *p* = 0.305. A significant difference between interaction partners was indicated for Responsiveness, F(2,104) = 5.93, *p* = 0.004, with follow-up Tukey multiple comparisons of means tests indicating that parents engage in responsiveness more than non-parents for both NPF (*p* = 0.007) and NPM (*p* = 0.007). A similar pattern of results emerged for Teaching, F(2,104) = 22.45, *p* < 0.001, with parents engaging in teaching more than non-parents for both NPF and NPM (*ps* < 0.001). However, the F tests for Encouragement showed a significant difference between partners, F(2,104) = 5.84, *p* = 0.003, but only between PF and NPM (*p* = 0.003). 

To further probe the difference in Encouragement between PF and NPF, a post hoc 3 (Interaction Partner) by 2 (Experience Level) Analysis of Variance (ANOVA) was conducted. Both interaction partner, F(2,104) = 22.45, *p* < 0.001, and experience level, F(2,104) = 22.45, *p* < 0.001, predicted encouragement, but no interaction effect was indicated F(2,104) = 22.45, *p* < 0.001. Tukey multiple comparisons of means tests indicate that NPM with low experience (M = 7.04, SD = 3.75) use encouragement less than PF with high experience (M = 11.30, SD = 2.85). This pattern of results suggests, again, that experiences with young children is a primary factor explaining adults use of encouragement during adult-child interactions. 

## 4. Discussion

The current study examined adult-child interactions with a toddler in a standardized playroom using parent females, non-parent females, and non-parent males. Our results suggest that both parenthood and adult gender impact aspects of adult-child interaction quality, quantity, and style. Overall, parent females showed more verbal engagement during interactions with a toddler. However, differences in experiences with young children seem to explain this effect. In contrast, the content of language directed toward the child differed systematically between the three groups compared here. Female parents used more nouns (i.e., labels) compared to non-parents. At the same time, female parents used fewer emotion words than non-parents-regardless of adult gender in both cases. Non-parent males asked more wh-questions during interactions, replicating previous work with fathers (i.e., male parents). Finally, examining engagement strategies during adult-child play revealed higher levels of responsiveness and teaching in parents regardless of gender. Together, the observed pattern of results shows influences of both adult gender and experiences of parenthood. 

Our results confirm previous findings showing differences in language quality between mothers and fathers, e.g., [12,20]. However, the current study extends these findings to males and females in general. Previous studies identified the unique roles of mothers and fathers for language [29], emotional [30], social [31], and cognitive development [32]. By studying undergraduate students during adult-child interactions with the same unfamiliar child, we removed the traditional roles of “motherhood” and “fatherhood” as confounding factors. Further, our analyses explicitly controlled for influences of experiences with young children. Therefore, our results make a compelling case for gender-specific interaction styles between adults and children that are not due to mere differences in experiences or parenting roles. By using language differently when interacting with children, males and females may offer unique contributions to a child’s development. Further, prior research also suggests that mothers’ and fathers’ role in fostering child development varies by domain. For example, parent-report measures on child development and the family environment indicate that mothers have a stronger impact on children’s language and cognitive development, whereas fathers may influence social and motor development more strongly [33]. These differences are not surprising, as mothers and fathers engage in different activities with their children [24]. The results reported here show that this pattern seems to hold for females vs. males in general and not just mothers vs. fathers [34].

When it comes to language content, the current study observed interesting findings that suggest that parents (here female parents in particular) may modify their language when interacting with children to focus on labeling objects and reduce complex concepts such as feelings or thoughts. This may be done to facilitate language learning and to avoid overwhelming the child with constructs that the child may not yet understand. Non-parent adults, in contrast, do not seem to follow this approach. This is evident by their increased use of emotion words. These results agree with research on child-directed speech that reports parents using exaggerated, slower, and simplified language when talking with a child. The current study did not examine adult speech for speed, tone, or overall complexity (especially grammatical complexity), but our observation that non-parents used more emotion words does agree with the concept of child-directed speech and a simplification of the language input. Parents used nouns to talk about objects that are physically present and can be seen and touched. In contrast, non-parents asked about feelings, wishes, and emotions–constructs that are not physically present and abstract. This is harder to understand for toddlers, suggesting that the non-parent adults did not adapt their language complexity to a level that is adequate for a toddler. 

The current findings do not identify any differences in language quantity or quality as a function of perceived child gender. Moreover, we did not observe any interactions between toddler gender and adult gender, suggesting no systematic differences between same-gender and mixed-gender adult-child dyads. This contrasts with prior findings of differences between fathers’ interactions with daughters as opposed to sons [35]. Similarly, others reported differences in how early childhood educators behave toward girls and boys [36]. The lack of significant differences in the current study could be driven by a combination of factors such as the age of the child, the context provided in our standardized playroom, and the nature of our adult participants (i.e., undergraduate students). Further, it is also possible that child gender was not salient to our adult participants as this was only manipulated by dressing the child differently. Nevertheless, the current results are encouraging by showing that girls and boys are engaged in similar ways by unfamiliar adults interacting with them for the first time.

### Limitations

Several limitations should be considered when interpreting the current results. For example, while the child was held constant in our design with non-parent adults, the age of the child did vary over the study period. Changes in the child’s age and familiarity with the testing room and situation may have implicitly influenced child behavior over the course of the study. However, we have not observed any such trends in our data and behavior of the child seemed more affected by randomly varying factors such as time of day and current state of the child (e.g., hungry, or sleepy). Another limitation is the lack of parent males and misbalance of male and female children in our parent sample. Unfortunately, lab closure due to the COVID-19 pandemic disrupted our study, resulting in an incomplete sample. Our results have important implications for gender differences in mother versus father language during parent–child interactions and should be explored further in future studies. An additional limitation in the parent sample is that parents played with their own child rather than an unknown child. This may have given parents an “advantage”. However, given this “advantage” it is noteworthy that we did not observe more differences between parent and non-parent adults in the current study. This suggests that adult-child play is similar for parents and non-parents. Further, our results with non-parents confirm previous studies with biological parents, again suggesting that behavior in the domains assessed here is surprisingly similar across parents and non-parents. Finally, there may have been a self-selection process in our undergraduate students who choose to participate in the current study. Unlike other studies, our study required an in-person visit to our lab and students who decide to complete in-lab studies may differ systematically from students wanting to complete only online studies.

## 5. Conclusions

In summary, the current study provides evidence for systematic differences between parents and non-parent female or male adults while interacting with young children. Female adults (regardless of parent-status) provide overall more language stimulation toward children. Further, content of language stimulation differs systematically between parents, male, and female adults. Male adults seem to ask children more questions during interactions (similar to what has been reported for fathers), whereas parents seem to use fewer emotion words and more nouns. By comparing non-parent adult male and female participants during interactions with the same child, our findings are the first to provide evidence that previously reported differences in parent–child interactions between mothers and fathers are not due to the child’s familiarity, experience with, or preference for one parent over the other. Therefore, our findings add support to the notion that mothers and fathers provide unique contributions to a child’s development and do not fulfill identical roles in the family system. At the same time, we also show that differences in how adults engage children may be more related to their own experiences, perceptions toward gender roles, and interests, rather than their biological gender.

## Figures and Tables

**Figure 1 children-09-01804-f001:**
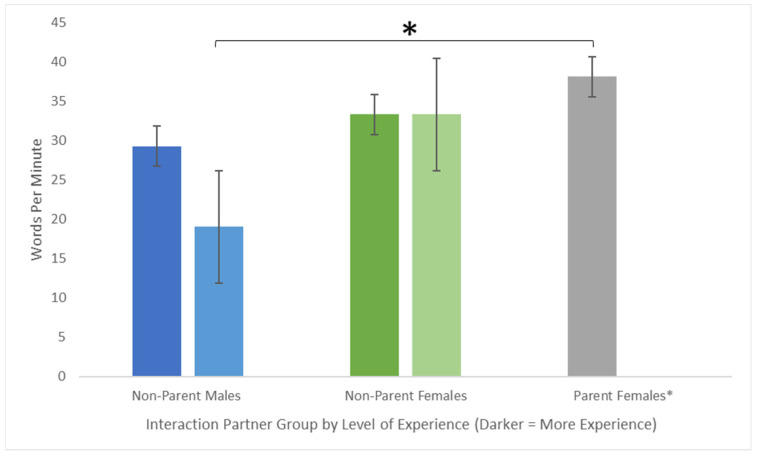
Differences in words produced per minute between interaction partner groups. Overall, words per minute differ between female parents and non-parent males only when accounting for experience with children. Note: * No Parent Females endorsed Low Experience with children.

**Figure 2 children-09-01804-f002:**
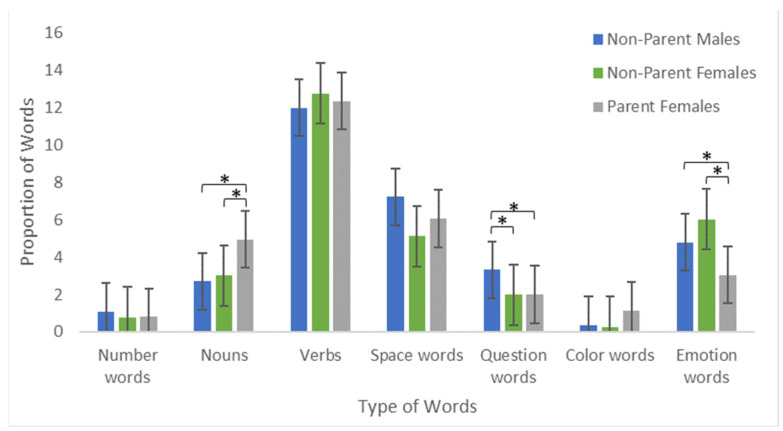
Differences in language quality (i.e., content) between interaction groups. Error bars represent Standard Errors. * *p* < 0.05.

**Figure 3 children-09-01804-f003:**
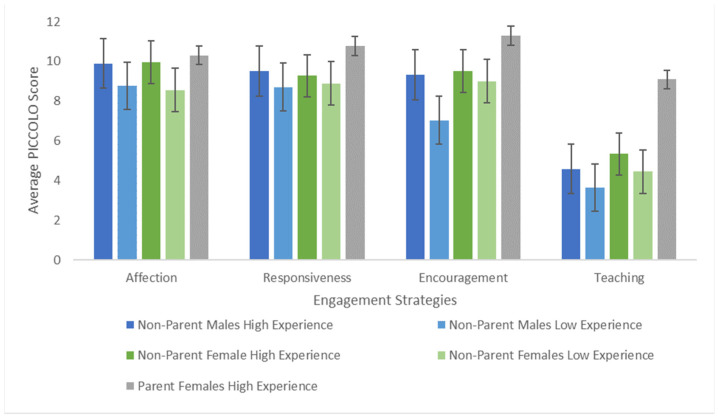
Differences in engagement strategies between interaction partner groups, accounting for experiences with children. Error bars represent Standard Errors. Overall, responsiveness and encouragement are employed more by parents, but employing encouragement is driven by experience rather than gender or parenthood status.

**Table 1 children-09-01804-t001:** Adult demographic information.

Gender	N	Race	Ethnicity
Non-Parent Female	41	White: 31 (76%) Black: 0 (0%) Asian: 8 (20%) Other: 1 (2%) No response: 1 (2%)	Hispanic: 1 (2%) Not Hispanic: 36 (88%) No response: 4 (10%)
Non-Parent Male	41	White: 26 (63%) Black: 7 (17%) Asian: 6 (15%) Other: 2 (5%) No response: 0 (0%)	Hispanic: 4 (10%) Not Hispanic: 35 (85%) No response: 2 (5%)
Parent Female *	24	White: 22 (92%) Black: 2 (8%) Asian: 0 (0%) Other: 0 (0%) No response: 0 (0%)	Hispanic: 4 (17%) Not Hispanic: 20 (83%) No response: 0 (0%)
Total	106	White: 79 (75%) Black: 9 (8%) Asian: 14 (13%) Other: 3 (3%) No response: 1 (1%)	Hispanic: 9 (8%) Not Hispanic: 91 (86%) No response: 6 (6%)
Campus average **	19,330	White: 70% Black: 5% Asian: 10% Other:4% No response: 1%	Hispanic: 4%
Pittsburgh average ***	303,160	White: 66.4% Black: 23.0% Asian: 5.8% Other: 0.3%	Hispanic: 3.4%

Note. * Parent Female demographic information based on parent report about child participant. ** Campus averages are taken from the National Center for Education Statistics (NCES) and based on Fall 2017 data. Due to differences in reported categories, campus averages do not sum to 100%. Data from the non-parent sample were collected between December 2018 and May 2019. *** City averages are taken from the U.S. Census Bureau and based on April 2020 data. Due to differences in reported categories, city averages do not sum to 100%. Data from the parent sample were collected between May 2019 and February 2020.

**Table 2 children-09-01804-t002:** Examples of word categories coded for language quality comparison.

Word Category	Examples
Noun *	cow, baby, daddy, car,…
Verb	play, do, can, feel,…
Number (excluding “one”)	zero, four, ten, double,…
Space word	narrow, up, left, inside,…
*Wh*-question *	what, where, who, why, how
Color	red, silver, gold, white,…
Emotion *	think, want, feel, like,…

Note. * indicates significant differences between interaction partner groups.

## Data Availability

Data is available upon request from the first author.

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
