# Peer review of "Influences of Adult Gender and Parenthood on Adult-Child Interaction Style"

_children, 2022, doi:10.3390/children9121804_

Round 1

Reviewer 1 Report

This is a highly interesting paper.  And although the lack of male parent particaption is a drawback (this is acknowledged), the results are intriguing in that they point to a particular sex-based contribution to parenting that has hitherto been elusive to pin down.  I am not a quanatative methodology person so cannot comment on the methods.

The paper ought to be read carefully to ensure that there is no ambiguity in describing findings that are unrelated to male parenting styles.   

Author Response

We thank the reviewer for their positive perception towards our manuscript and our research study. Based on the reviewer’s feedback, we have spell-checked the entire manuscript using proofing tools.

Regarding the lack of a male parent group, we completely agree with the reviewer. This group was initially planned, but could not be completed due to the onset of the Covid-19 pandemic. We have re-worded parts of the manuscript to avoid the ambiguity described by the reviewer. However, if any concerns persist, we would appreciate hearing more recommendations for changes in our wording and how we interpret and describe our findings.

Thank you again and we hope you will agree that our revised manuscript addresses the concerns raised in the review.

Reviewer 2 Report

Thank you firstly for the opportunity to conduct a review of this work.

The proposed approach is interesting and may have positive repercussions in increasing knowledge regarding the characteristics of adult-child interactions.

The conceptual framework is outlined comprehensively, with rich bibliographical references.

The findings and conclusions are reported correctly and accurately, and they contribute to the increase of knowledge on the topic.

Minor reviews are suggested:

- l. 107: it should be specified whether ethical standards defined by relevant field boards (e.g., APA) were applied;

- How was the location in which the observations were conducted arranged? The setting prepared should be described, with reasons for the choices made;

- with regard to the EPIQ instrument, the results regarding validity should be reported in more detail.

Author Response

We thank the reviewer for their positive perception towards our manuscript and our research study. The reviewer has provided us with several comments and suggestions for revisions. Specifically:

1) it should be specified whether ethical standards defined by relevant field boards (e.g., APA) were applied

Response: Our research study has been approved by an Institutional Review Board (IRB) and followed established ethical standards for behavioral science procedures. This information is now highlighted in our revised document. Please see page 3 in the revised manuscript.

2) How was the location in which the observations were conducted arranged? The setting prepared should be described, with reasons for the choices made.

Response: We thank the reviewer for pointing out this oversight. We now clarify the room, camera, and toy selection in more detail. Please see page 5 in the revised manuscript.

3) with regard to the EPIQ instrument, the results regarding validity should be reported in more detail.

Response: We thank the reviewer for this excellent suggestion. Additional details regarding the EPIQ are now provided on pages 5 and 6 of the revised manuscript. However, if the reviewer has suggestions for additional information to include, please do let us know. The entire EPIQ will be available as supplemental material with the submission.

Thank you again and we hope you will agree that our revised manuscript addresses the concerns raised in the review.